# DragLoRA: Online Optimization of LoRA Adapters for Drag-based Image Editing in Diffusion Model

**Siwei Xia** [1]  **Li Sun** [1 2]  **Tiantian Sun** [1]  **Qingli Li** [1]

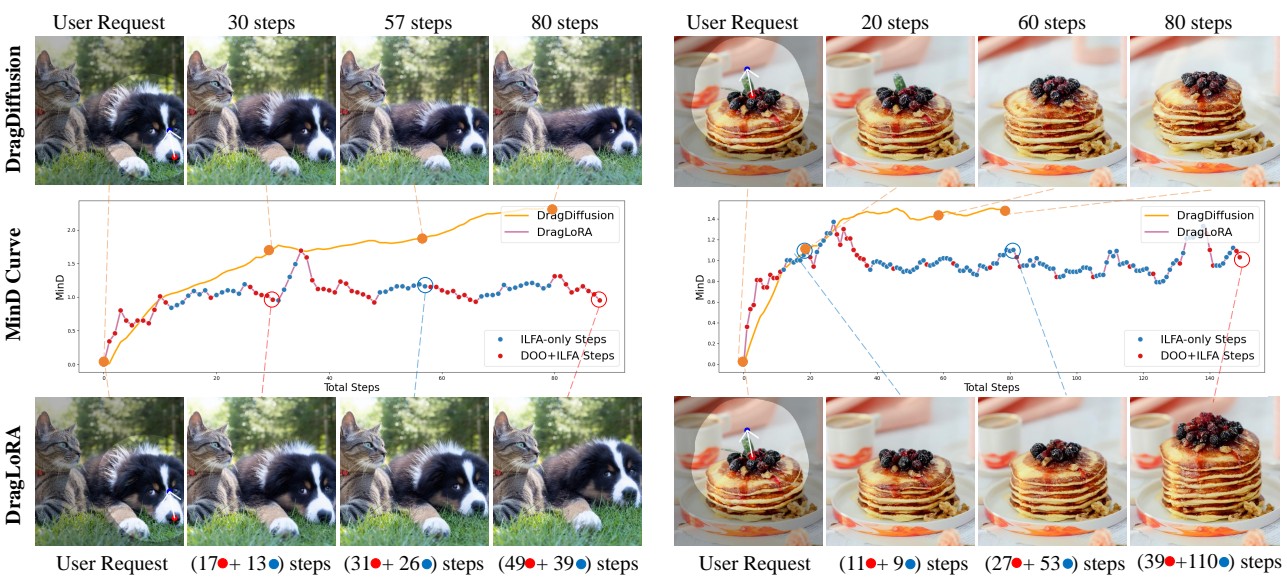

*Figure 1.* Visual comparison between DragDiffusion (Shi et al., 2024b) and DragLoRA at each step. For a given image and user request, we present dragged images at four intermediate steps. In DragLoRA, the steps are divided into two types: DOO+ILFA (Red) or ILFA-only (Blue), where DOO stands for Dual-Objective Optimization and ILFA stands for Input Latent Feature Adaptation. DragDiffusion requires 80 optimization steps but produces results with lower fidelity. In contrast, DragLoRA achieves more precise deformations with fewer optimization steps and more total steps, which consumes less time thanks to the high-efficiency of ILFA. The middle $minD$ curves demonstrate that DragLoRA achieves better point tracking, preserving source details more accurately than DragDiffusion.

## Abstract

Drag-based editing within pretrained diffusion model provides a precise and flexible way to manipulate foreground objects. Traditional methods optimize the input feature obtained from DDIM inversion directly, adjusting them iteratively to guide handle points towards target locations. However, these approaches often suffer from limited accuracy due to the low representation ability of the feature in motion supervision, as well as inefficiencies caused by the large search space required for point tracking. To address these limitations, we present DragLoRA, a novel framework that integrates LoRA (Low-Rank Adaptation) adapters into the drag-based editing pipeline. To enhance the training of LoRA adapters, we introduce an additional denoising score distillation loss which regularizes the online model by aligning its output with that of the original model. Additionally, we improve the consistency of motion supervision by adapting the input features using the updated LoRA, giving a more stable and accurate input feature for subsequent operations. Building on this, we design an adaptive optimization scheme that dynamically toggles between two modes, prioritizing efficiency without compromising precision. Extensive experiments demonstrate

[1]Shanghai Key Laboratory of Multidimensional Information Processing. [2]Key Laboratory of Advanced Theory and Application in Statistics and Data Science, East China Normal University, Shanghai, China. Correspondence to: Li Sun <sunli@ee.ecnu.edu.cn>.

*Proceedings of the $42^{nd}$ International Conference on Machine Learning*, Vancouver, Canada. PMLR 267, 2025. Copyright 2025 by the author(s).

that DragLoRA significantly enhances the control precision and computational efficiency for drag-based image editing. The Codes are available at: https://github.com/Sylvie-X/DragLoRA.

## 1. Introduction

The Stable Diffusion (SD) model has demonstrated remarkable capabilities in synthesizing high-quality images from textual prompts. Based on the pre-trained SD model, many works (Hertz et al., 2022; Zhang et al., 2023; Hertz et al., 2023) have sought to enhance control over the generated images, often relying on detailed text prompts or reference images to specify generation conditions. While these methods offer various types of editing, they typically require users to provide complex instructions to describe their desired outputs, which can be cumbersome and restrictive in scenarios where only minimal modification is required.

Recent advances in drag-based image editing enable intuitive point-driven manipulation within pre-trained generative models. By specifying pairs of source and target points, users can interactively guide object deformation, iteratively "dragging" content from the source location toward the target. These methods typically operate through two sequential stages: motion supervision, which computes directional gradients to align features with the desired movement, and point tracking, which updates handle positions based on the evolving feature space. While this paradigm reduces reliance on complex textual or reference inputs, existing approaches often struggle with precision and efficiency. Direct optimization of latent features, such as DDIM-inverted representations, introduces instability due to limited feature expressiveness in motion supervision, while the iterative tracking process incurs high computational costs from searching large spatial regions.

In this work, we address these limitations by augmenting the SD model with DragLoRA, a dynamically optimized adapter integrated into all attention layers of the pre-trained Unet. Unlike prior methods that directly optimize latent features, our method performs online adaptive learning during drag-based editing, enabling adjustments of LoRA parameters tailored to user interactions. The expanded optimization space enhances the model's capacity to represent foreground deformations, allowing precise alignment of handle points with target positions at each motion supervision step. By decoupling deformation control from static latent representations, DragLoRA mitigates the instability caused by limited feature expressiveness, hence reducing reliance on iterative large-scale searches.

However, we observe that unrestricted LoRA optimization guided solely by drag loss can lead to deviations from the original image. To address this, we introduce a delta de-

noising score (DDS) loss to regularize the online training. Specifically, DragLoRA first predicts a clean feature from the DDIM-inverted input at timestep of $t$, which is then perturbed with noise at a randomly sampled timestep $t'$. The DDS loss is computed as the difference between the noise predictions of the original UNet and the DragLoRA enhanced UNet for the perturbed feature. By jointly minimizing the extra loss with the original motion supervision objective, our method preserves semantic fidelity while enabling flexible deformations. Furthermore, to ensure motion consistency across iterative edits, we adapt the input feature to accumulate deformation effects. At each step, the input feature is denoised from $t$ to $t-1$ using DragLoRA's predicted noise, then re-noised back to $t$ with random perturbations, for the next optimization. This cycle progressively aligns the feature with the accumulated deformation trajectory, propagating handle point adjustments into the latent space and stabilizing motion supervision through coherent feature updates.

In practice, we observe that handle points can be driven toward target positions through input adaptation alone, even without explicit motion supervision. This occurs because accumulated gradients from previous optimizations can be utilized for moving handle points at the new positions without extra driving force. In each gradient step, although the specific tasks are not exactly the same, they share a low-variance handle feature and a common direction. DragLoRA can learn these commonalities and generalize, which is comparable with meta-learning. To leverage this, we employ an adaptive optimization strategy: when point tracking achieves sufficient quality, LoRA updates are bypassed to prioritize efficiency. Conversely, if tracking deviates (e.g., due to occlusions or ambiguous textures), motion supervision is triggered to refine the LoRA parameters, ensuring robust deformation control. By dynamically toggling between motion supervision and input adaptation, DragLoRA enables efficient handle localization with minimal optimization steps, as it selectively optimizes LoRA only when necessary.

The contributions of this paper lie in following aspects.

- We propose DragLoRA, a parameterized adapter enables online optimization following user's interactions. By replacing direct latent feature optimization with dynamic model adaptation, DragLoRA enhances fine-grained deformation capacity while preserving the pre-trained diffusion priors.

- We introduce a dual-objective framework that combines drag loss with a DDS loss, computed by comparing noise predictions of the original and LoRA-augmented Unets on perturbed features. Coupled with a cyclic denoise-renoise process, which iteratively propagates handle point adjustments into the latent

space, this framework ensures semantic fidelity with the source image and stabilizes motion supervision through accumulated deformation trajectories.

- We design an adaptive optimization strategy that dynamically toggles between two modes. By evaluating tracking quality, DragLoRA prioritizes efficient input feature updates when tracking succeeds and activates motion supervision for LoRA refinement when deviations occur, minimizing redundant optimization steps.

## 2. Related Works

**Applications of diffusion model on image editing.** Diffusion models (Ho et al., 2020; Song et al., 2020b; Dhariwal & Nichol, 2021) are initially designed for iterative denoising in the pixel domain and later accelerated by operating in the latent feature space (Rombach et al., 2022). Trained on large-scale image-text datasets, Stable Diffusion can generate high-quality images conditioned on input text. These models enable various image editing tasks with little to no additional training. Typically, the source image is noised to an intermediate timestep suitable for editing, then denoised back with target conditions to modify the content. SDEdit (Meng et al., 2022) applies random noise directly following the DDPM schedule, while DDIM inversion (Song et al., 2020a) and its advanced versions (Mokady et al., 2023; Miyake et al., 2023) have been shown to better preserve details from the source image.

Among various editing techniques, text-based editing is one of the most explored. P2P (Hertz et al., 2022) achieves precise modifications by adjusting the cross-attention matrix in the UNet, while PnP (Tumanyan et al., 2023) and Masactrl (Cao et al., 2023) focus on modifying the self-attention layers. Optimization-based methods, on the other hand, are typically trained on a single image (Kawar et al., 2023; Valevski et al., 2023; Hertz et al., 2023) or a small set of images (Gal et al., 2022; Ruiz et al., 2023; Kumari et al., 2023), allowing the model to learn detailed appearances of the editing target and adapt to arbitrary text prompts. In contrast, reference-based editing directly feeds a reference image into the model, often requiring a large dataset for effective learning (Wei et al., 2023; Ye et al., 2023). The reference can take various forms, including a standard image or structural information extracted from the target image (Zhang et al., 2023).

**Dragged-based image editing** has gained significant attention for its intuitive approach to modifying images through user-defined handle and target points. DragGAN (Pan et al., 2023) is the first to demonstrate the feasibility of "dragging" in pre-trained StyleGAN models, introducing motion supervision and point tracking mechanisms. Subsequent advancements have expanded this concept to diffusion mod-

els. DragDiffusion (Shi et al., 2024b) adapts the point-based dragging technique to the SD model, enhancing generative quality and realizing precise spatial control. SDE-Drag (Nie et al., 2024) presents a unified probabilistic formulation for diffusion-based image editing, including dragging. DragNoise (Liu et al., 2024) utilizes U-Net's noise predictions for efficient point-based editing while maintaining semantic coherence. FreeDrag (Ling et al., 2024) improves stability and efficiency by narrowing the search region for handle points and incorporating adaptive feature updates. EasyDrag (Hou et al., 2024) simplifies the user interaction process, making image editing more intuitive and accessible. GoodDrag (Zhang et al., 2024) introduces an alternating drag and denoising framework, improving result fidelity and reducing distortion. StableDrag (Cui et al., 2024) addresses challenges in point tracking and motion supervision by developing a discriminative point tracking method and a confidence-based latent enhancement strategy, resulting in more stable and precise drag-based editing. AdaptiveDrag (Chen et al., 2024) proposes a mask-free point-based editing approach, leveraging super-pixel segmentation for adaptive steps. ClipDrag (Jiang et al., 2024) leverages CLIP for text-guided editing, offering semantic control over image content, while DragText (Choi et al., 2024) facilitates text-guided drag by optimizing text embeddings alongside image features. FastDrag (Zhao et al., 2024) enabled quick image modifications without the need for iterative optimization. GDrag (Lin et al., 2025) categorizes point-based manipulations into three atomic tasks with dense trajectory, achieving less ambiguous outputs.

Besides point-based drag editing, DragonDiffusion (Mou et al., 2023), DiffEditor (Mou et al., 2024), and RegionDrag (Lu et al., 2024) extend drag editing to regions or incorporate a reference image for editing. Additionally, InstantDrag (Shin et al., 2024) and LightningDrag (Shi et al., 2024a) train general-purpose models for drag-based editing, enabling rapid adaptation across various tasks and datasets.

The proposed DragLoRA is a point-based drag editing method. Compared to other methods in the same category, it achieves state-of-the-art performance on key metrics while reducing time costs. In contrast to general models like LightningDrag, DragLoRA avoids the heavy burden of offline training and delivers better results.

## 3. Method

The proposed DragLoRA framework, as is shown in Figure 2, augments the pretrained Stable Diffusion (SD) model with a dynamically optimized adapter to enable interactive drag-based image editing. Our method has two modes for driving handle points, which are LoRA training with motion supervision and input feature adaptation without it. And we design an adaptive scheme to combine them for drag-based

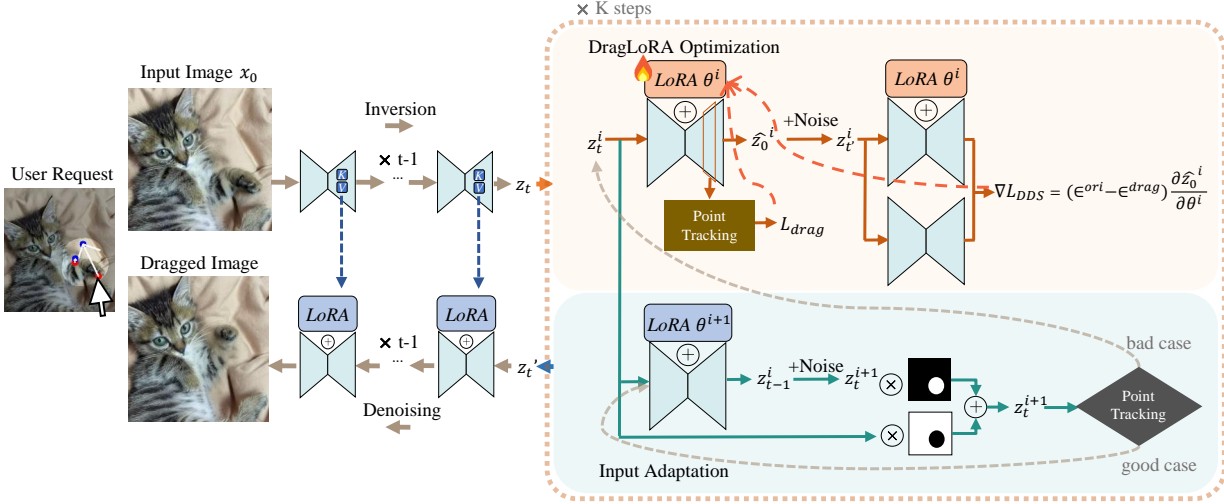

*Figure 2.* Overview of our proposed DragLoRA. Given an inversion code $z_t$ at $t = 35$ from a source image $x_0$, we incorporate a LoRA adapter and optimize it online using $L_{\text{drag}}$ and $L_{\text{DDS}}$. $L_{\text{drag}}$ primarily guides handle point movement, while $L_{\text{DDS}}$ constrains LoRA to remain close to the original model, preserving the fidelity of the edited image. Additionally, the input feature to the UNet undergoes a denoise-renoise cycle in the foreground region, allowing the LoRA adapter to drive handle points even without gradient updates. DragLoRA dynamically switches between motion supervision and input-adaptation-only modes based on point tracking quality, ensuring both efficiency and stability in the editing process.

editing.

### 3.1. Preliminaries: Diffusion Model and Drag-based Image Editing

**Diffusion models** ([Ho et al., 2020](#); [Rombach et al., 2022](#)) have the forward process, in which an input image or latent representation $z_0$ is progressively perturbed by Gaussian noise over $T$ timesteps:

$$z_t = \sqrt{\alpha_t}\, z_{t-1} + \sqrt{1-\alpha_t}\, \epsilon_t$$
$$= \sqrt{\bar{\alpha}_t}\, z_0 + \sqrt{1-\bar{\alpha}_t}\, \epsilon, \quad \epsilon_t, \epsilon \sim \mathcal{N}(0, I), \tag{1}$$

where $\alpha_t = 1 - \beta_t$ and $\bar{\alpha}_t = \prod_{s=1}^{t}(1-\beta_s)$ is the cumulative noise retention coefficient, and $\{\beta_t\}$ defines a fixed noise schedule. As $t \to T$, $\bar{\alpha}_t$ decays to zero, and $z_t$ becomes a pure noise. The reverse process aims to recover $z_0$ by iteratively denoising $z_t$. Denoising Diffusion Implicit Models (DDIM) ([Song et al., 2020a](#)) propose a non-Markovian sampling process:

$$z_{t-1} = \sqrt{\bar{\alpha}_{t-1}}\left(\frac{z_t - \sqrt{1-\bar{\alpha}_t}\,\epsilon_\theta(z_t, t)}{\sqrt{\bar{\alpha}_t}}\right) + \sqrt{1-\bar{\alpha}_{t-1}}\cdot\epsilon_\theta(z_t, t), \tag{2}$$

giving an accelerated generation. Crucially, DDIM supports deterministic inversion: given $z_0$, the noisy latent $z_t$ at any $t$ can be reconstructed via:

$$z_{t+1} = \sqrt{\bar{\alpha}_{t+1}}\left(\frac{z_t - \sqrt{1-\bar{\alpha}_t}\,\epsilon_\theta(z_t, t)}{\sqrt{\bar{\alpha}_t}}\right) + \sqrt{1-\bar{\alpha}_{t+1}}\,\epsilon_\theta(z_t, t). \tag{3}$$

This inversion maps $z_0$ to an editable latent at time $t$ for different tasks, *e.g.*, $z_{35}$ for drag editing.

**Drag-based image editing** is typically performed in a pre-trained generative model, such as GAN or diffusion-based model. In this task, user provides a set of source points $\mathbf{p}_i$ and corresponding target points $\mathbf{g}_i$, where $\mathbf{p}_i, \mathbf{g}_i \in \mathbb{R}^2$ represent 2D pixel coordinates in the image plane. For SD, a LoRA adapter is first fine-tuned on the input image to ensure that the edited results retain a high similarity to the original. The latent feature, $z_{35}^0$ at timestep $t = 35$, is then obtained via DDIM inversion and serves as the optimization target. During editing, two stages alternate:

**Motion Supervision**: A gradient-based objective adjusts $z_{35}$ to morph the neighborhood of temporal target points $\mathbf{h}_i + \mathbf{d}_i$ into the corresponding regions around handle points $\mathbf{h}_i$, using features from a selected UNet layer to compute directional guidance, as is shown in (4).

$$L_{\text{drag}} = \sum_{i=1}^{N} \|\text{sg}(F(z_{35}^0, \mathbf{h}_i^0)) - F(z_{35}, \mathbf{h}_i + \mathbf{d}_i)\|_1 \tag{4}$$

where $F(\cdot)$ extracts features from the specified UNet layer and $\text{sg}(\cdot)$ detaches the possible gradients. $N$ denotes total number of handle points specified by user. $\mathbf{d}_i$ is the normalized displacement vectors $\mathbf{d}_i = \frac{\mathbf{g}_i - \mathbf{h}_i}{\|\mathbf{g}_i - \mathbf{h}_i\|_2}$. To mitigate cumulative errors, we adopt the initial latent input $z_{35}^0$ and handle point $\mathbf{h}_i^0 = \mathbf{p}_i$ to obtain fixed target features, which

is different from (Pan et al., 2023; Shi et al., 2024b). To preserve the original content outside the edited regions, an optional constraint $L_{\text{Mask}}$ is applied:

$$L_{\text{Mask}} = \|(z_{34} - z_{34}^0) \cdot (1 - M)\|_1, \qquad (5)$$

where $z_{34}$ is denoised from $z_{35}$ by (2) through a forward pass of the model, and $M$ is a given binary mask indicating editable regions. This ensures that non-target areas remain unchanged during optimization.

**Point Tracking**: The updated $\hat{z}_{35}$ is reprocessed through the UNet to locate the new handle positions $\mathbf{h}_i^{k+1}$, which serve as new guidance for subsequent motion supervision. Here, $\Omega(\mathbf{h}_i^k, r_2)$ defines a rectangular search area centered at the previous handle point $\mathbf{h}_i^k$, with $r_2$ controlling its size.

$$\mathbf{h}_i^{k+1} = \arg \min_{\mathbf{q} \in \Omega(\mathbf{h}_i^k, r_2)} \left\| F(\hat{z}_{35}, \mathbf{q}) - F(z_{35}^0, \mathbf{h}_i^0) \right\|_1 \quad (6)$$

Additionally, the new handle point $\mathbf{h}_i^{k+1}$ can be used to assess the quality of drag. We compute the best matching distance between $\mathbf{h}_i^{k+1}$ and $\mathbf{h}_i^0$, as defined in (7). A lower value of $minD$ indicates higher confidence in point tracking and the success of the previous optimization for motion supervision.

$$minD = \left\| F(\hat{z}_{35}, \mathbf{h}_i^{k+1}) - F(z_{35}^0, \mathbf{h}_i^0) \right\|_1 \qquad (7)$$

Furthermore, we evaluate a geometric metric, the Euclidean distance $d(\mathbf{h}_i^{k+1}, \mathbf{h}_i^k + \mathbf{d}_i)$, between $\mathbf{h}_i^{k+1}$ and $\mathbf{h}_i^k + \mathbf{d}_i$. Intuitively, this distance should be small, as motion supervision requires $F(z_{35}, \mathbf{h}_i^k + \mathbf{d}_i)$ to align closely with $F(z_{35}^0, \mathbf{h}_i^0)$ from the previous optimization step. Thus, the tracked point should not deviate significantly from it. These additional metrics are not used in previous methods and are specific to our proposed online adaptive optimization strategy. Further details can be found in Section 3.4.

## 3.2. DragLoRA and Its Online Optimization

Building on existing works, we introduce DragLoRA, a novel framework that improves the precision and efficiency of user-guided deformations through online optimization of LoRA adapters. Instead of directly optimizing latent feature $z_{35}$, DragLoRA dynamically adjusts the parameters $\Delta\theta$ of LoRA, which are integrated into the UNet parameterized by $\theta$. This approach increases the capacity of the model and decouples deformation control from latent features, giving fine-grained adjustments while preserving semantic fidelity. To minimize additional computation, we initialize the LoRA with the weights from reconstruction fine-tuning on the input image, without introducing extra LoRA modules. This

makes the model size unchanged compared to (Shi et al., 2024b).

However, we find that optimizing LoRA exclusively for drag-based editing leads to performance degradation, as iterative fine-tuning causes the LoRA-enhanced model to deviate significantly from the original pretrained model. To address this, we propose a dual-objective optimization scheme, combining the drag loss $L_{\text{drag}}$ with a delta denoising score (DDS) loss $L_{\text{DDS}}$ (Hertz et al., 2023; Arar et al., 2024). Specifically, we first use DragLoRA to predict the clean signal $\hat{z}_0$ based on the feature $z_{35}$ from DDIM inversion. The forward process is then applied, adding noise according to (1), transforming $\hat{z}_0$ into $\hat{z}_{t'}$, where $t'$ is a random timestep. Finally, $\hat{z}_{t'}$ is passed through both the LoRA-enhanced and pretrained models to compute the discrepancy between their noise predictions $\epsilon^{drag}$ and $\epsilon^{ori}$. The gradient of the DDS loss is computed as follows:

$$\nabla_{\Delta\theta} L_{\text{DDS}} = (\epsilon^{ori} - \epsilon^{drag}) \frac{\partial \hat{z}_0}{\partial \Delta\theta} \qquad (8)$$

Note that this gradient only takes effect through $\hat{z}_0$. The total loss can be expressed as $L = L_{\text{drag}} + \lambda_{Mask} L_{\text{Mask}} + \lambda_{DDS} L_{\text{DDS}}$. While the drag loss aligns handle points with their target positions, the additional gradient ensures consistency with the pretrained model. This dual-objective optimization (DOO) effectively balances precise deformation control with fidelity to the original model, mitigating the instability introduced by unrestricted LoRA optimization.

## 3.3. Input Latent Feature Adaptation (ILFA)

To enhance the stability of motion supervision, we introduce a cyclic denoise-renoise process that adapts the input latent feature for drag-based editing. At each iteration, the input feature $z_{35}$ at timestep $t = 35$ is first denoised to $t - 1$ using DragLoRA's predictions, then re-noised back to $t$ with random Gaussian noise. This cycle propagates handle point adjustments into the latent feature space, ensuring coherent updates on LoRA parameters across iterations.

For denoising, we use the full model, including LoRA, to perform one step of DDIM denoising as defined in (2). Although DDIM can also be used for re-noising, we find that it produces inferior results compared to the DDPM schedule defined in (1). Moreover, this scheme is only carried out within the foreground mask, and background region keeps untouched. The proposed ILFA scheme, combined with the dual-objective loss, achieves robust and stable optimization, effectively addressing the challenges of unrestricted LoRA fine-tuning.

## 3.4. Adaptive Optimization Scheme with Two Modes

The input adaptation scheme works effectively with motion supervision. In some cases, it can drive the handle points

**Algorithm 1** Drag Updates

**Input:** inverted latent code $z_t$, mask $M$, unet $\theta$, reconstruction lora $\Delta\theta_{rec}$, source points $p$, target points $g$
**Initialize** $\Delta\theta = \Delta\theta_{rec}$, $z_t^0 = z_t$, $h = p$, $k = 0$
$F^0 = $ApplyModel$(\theta + \Delta\theta, z_t^0)$
**while** $k < K$ and $||h - g||_2 > l_1$ **do**
  $F = $ApplyModel$(\theta + \Delta\theta, z_t)$
  $h, minD = $PointTrack$(F, F^0, p, g, h)$
  **if** $k > k_{ini}$ and $minD < d_1$ and $||h - n||_2 < l_2$ **then**
    **while** $minD < d_2$ and $||h - g||_2 > l_1$ **do**
      Adapt $z_t$
      $F = $ApplyModel$(\theta + \Delta\theta, z_t)$
      $h, minD = $PointTrack$(F, F^0, p, g, h)$
    **end while**
  **else**
    $d = \frac{g - h}{||g - h||_2}$
    $n = h + d$
    Optimize $\Delta\theta$ by losses defined as Eq.(4)(5)(8)
    Adapt $z_t$
    $k+ = 1$
  **end if**
**end while**

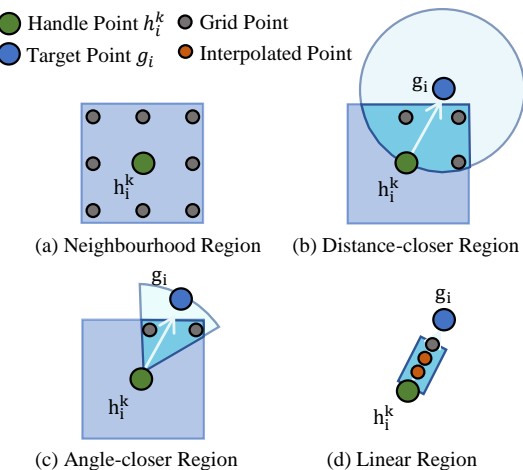

*Figure 3.* Comparison among different point tracking schemes. (a) A common strategy locates new point in a square neighborhood around current handle point. (b) and (c) reduces the search region using target point, only grid points in the intersection region are considered as candidate. (d) uses a linear line and needs feature interpolation. (b) and (d) are initially proposed in (Jiang et al., 2024) and (Ling et al., 2024), respectively.

toward their targets even without LoRA updates. This is because DragLoRA learns to move the handle points towards the desired direction through previous optimization steps, reducing the need for further LoRA adjustments. This strategy has high efficiency, as it does not need backpropagation.

To balance efficiency and robustness, we propose an adaptive switching scheme (ASS) between two modes, DOO plus ILFA and ILFA-only, based on the quality of point tracking. In the ILFA-only mode, DragLoRA updates the latent feature to guide the handle points to the target positions. When point tracking locates a confident handle point with a small enough $minD$ defined in (7), and is not far from the temporal target $\mathbf{h} + \mathbf{d}$, ILFA-only is activated. However, in challenging situations such as occlusions or texture ambiguities, point tracking may degrade, requiring further refinement. In these cases, DOO plus ILFA begins so that a gradient-based objective is used to adjust LoRA parameters and stabilize the deformation. This ensures that the handle points remain accurately positioned even in difficult scenarios. The proposed ASS dynamically toggles between these two modes based on point tracking quality. This adaptive scheme makes DragLoRA to efficiently handle a variety of scenarios while maintaining robustness. Details about it are provided in Algorithm 1.

### 3.5. Efficient Point Tracking (EPT)

To further enhance the efficiency of drag editing, we investigate different point tracking strategies for handle points, as shown in Figure 3. We find that DragLoRA can effectively

reduce the search region, and it outperforms traditional methods in terms of efficiency. Unlike conventional neighborhood searching, the distance-closer (Jiang et al., 2024) and angle-closer regions constrain the candidate points for handle tracking, preventing unnecessary location reversion. Specifically, the target or current handle point serves as the center of a circle, with their distance defining the radius. Only the grid points within the intersection of the circle (or sector) and the neighboring region are considered as candidates for comparison with the original source point feature. A more aggressive search strategy, proposed in (Ling et al., 2024), follows a straight line from the handle to the target point. In DragLoRA, we choose the distance-closer and angle-closer strategies for point tracking, as the former provides the best performance and the latter is the most efficient. Additionally, to prevent insufficient optimization due to rapid advancement of point coordinates, we determine whether to proceed or retain the previous handle point based on the minD. Further details are provided in Appendix B.3.

## 4. Experiments

### 4.1. Implementation Details

We use Stable Diffusion 1.5 (Rombach et al., 2022) as the base model. Following DragDiffusion (Shi et al., 2024b), we train reconstruction LoRA for each image in 80 steps with a learning rate of 0.0005. Given a total of 50 timesteps,

we optimize DragLoRA at $t = 35$ with a learning rate of 0.0001. The ranks of both LoRAs are set as 16, and DragLoRA is initialized from RecLoRA. We employ the Adam optimizer, and set $\lambda_{Mask} = 0.1, \lambda_{DDS} = 50, K = 80, k_{ini} = 10, l_1 = 1, l_2 = 1.4, d_1 = 1, d_2 = 1.3$. In terms of EPT, DragLoRA primarily utilizes the distance-closer region. To speed up the dragging process, the angle-closer region is used in DragLoRA-Fast. After drag updates, we apply DragLoRA to all the remaining timesteps to enhance the drag effects. If not specified otherwise, all of our experiments are conducted on a single NVIDIA RTX 4090 GPU.

## 4.2. Qualitative Evaluation

To validate the effectiveness of our proposed DragLoRA, we conduct extensive experiments on DragBench (Shi et al., 2024b), Drag100 (Zhang et al., 2024), VITON-HD(Choi et al., 2021) and private collections. We compare DragLoRA qualitatively with existing drag methods: DragDiffusion (Shi et al., 2024b), DragNoise (Liu et al., 2024), GoodDrag (Zhang et al., 2024). The visual results are presented in Figure 4. Our method and GoodDrag both demonstrate superior editability, while DragDiffusion and DragNoise fail to achieve the target specified by the input annotations, such as not being able to close the duck's mouth (third row). Additionally, compared to GoodDrag, our DragLoRA maintains higher fidelity. For example, after moving the camera, our method generates a more natural-looking face (first row).

## 4.3. Quantitative Evaluation

To further compare with more methods (Cui et al., 2024; Hou et al., 2024; Ling et al., 2024; Choi et al., 2024; Chen et al., 2024; Jiang et al., 2024; Lin et al., 2025; Shin et al., 2024; Shi et al., 2024a; Nie et al., 2024; Zhao et al., 2024), we conduct metric evaluations on DragBench. We use publicly available code from other works, except for ClipDrag (Jiang et al., 2024) and GDrag(Lin et al., 2025), which has not open-sourced its code, and categorize all drag methods into three types: optimization-based (Optim), encoder-based(Enc), and training-free (TrFree). We test DragText (Choi et al., 2024) based on DragDiffusion (Shi et al., 2024b). We evaluate the number of trainable parameters, the steps required to pre-train the reconstruction LoRA, the maximum step set for online optimization, and the associated time. Note that since the reconstruction LoRA can be trained offline and reused for multiple drag editings by various annotations of points, the time spent on this process is not included, which is about 48s per image over 80 steps on one NVIDIA 4090 GPU. As EasyDrag (Hou et al., 2024) requires more computational resources, we individually evaluate it on one NVIDIA A40 GPU.

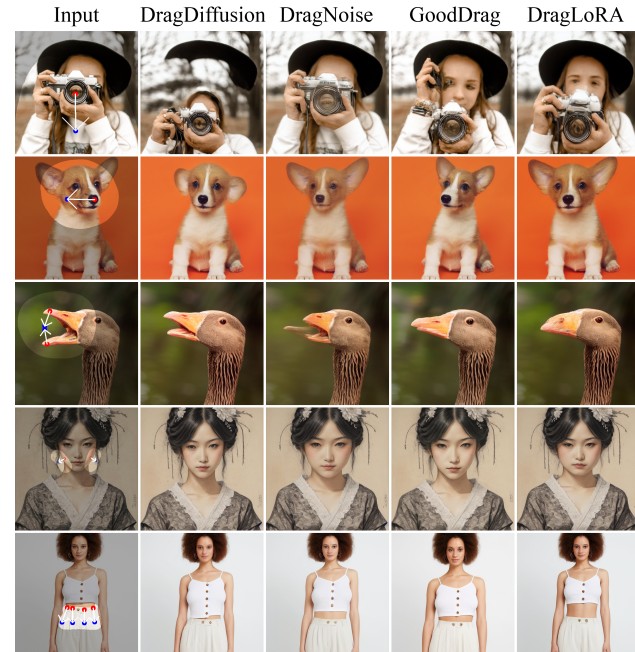

Input    DragDiffusion   DragNoise    GoodDrag    DragLoRA

*Figure 4.* Qualitative comparisons with (Shi et al., 2024b; Liu et al., 2024; Zhang et al., 2024). The proposed DragLoRA outperforms existing approaches in both perceptual quality and the accuracy of drag editing.

**Evaluation Metrics**: Following (Shi et al., 2024b), we adopt 1-LPIPS (Zhang et al., 2018) and MD (Mean Distance) to assess consistency with the original image and editing accuracy, respectively. The former calculates 1 minus feature difference between the original and edited images using AlexNet (Krizhevsky et al., 2012), while the latter computes coordinate discrepancy between handle points in the edited image and target points using DIFT (Tang et al., 2023). However, We find that using DIFT to search point coordinates across the entire image may not be accurate, leading to unreliable metrics. Similar to (Lu et al., 2024), we introduce m-MD (masked-MD), which restricts the DIFT search into the edit region specified by input mask $M$, thereby reducing uncertain errors. While m-MD is numerically lower than MD, it may yield optimistic scores in cases of image distortion, where the mask constraint forces final handle points closer to the target. Thus, the two metrics complement each other. As demonstrated in Table 1, DragLoRA achieves the best MD and competitive m-MD, with enhanced efficiency, compared to the state-of-the-art method GoodDrag. And our DragLoRA-Fast, which adopts angle-closer EPT, demonstrates significant time efficiency among optimization-based approaches while maintaining strong editability.

**Drag-Back Evaluation**: As the extent of editing increases, 1-LPIPS naturally decreases. Especially in cases of signifi-

*Table 1.* Quantitative comparisons on DragBench. **Bold**: best, underline: second best, *: not open-sourced. DragLoRA adopts distance-closer EPT while DragLoRA-Fast utilizes angle-closer EPT. Our methods demonstrate state-of-the-art editing quality and the lowest time consumption among optimization-based approaches.

| METHODS | 1-LPIPS ↑ | MD ↓ | M-MD ↓ | TIME(S) ↓ | PARAMS(M) | RECSTEPS | DRAGSTEPS | CATEGORY |
|---|---|---|---|---|---|---|---|---|
| DRAGDIFFUSION | 0.88 | 32.13 | 30.71 | 32.93 | 0.07 | 80 | 80 | OPTIM |
| DRAGNOISE | 0.89 | 35.17 | 30.66 | 30.53 | 0.33 | 200 | 80 | OPTIM |
| STABLEDRAG | **0.91** | 36.46 | 35.88 | 37.21 | 0.07 | 80 | 80 | OPTIM |
| EASYDRAG | **0.91** | 38.28 | 37.67 | 44.69 | 0.07 | 0 | 80 | OPTIM |
| GOODDRAG | 0.87 | 24.26 | **21.86** | 56.97 | 0.07 | 70 | 210 | OPTIM |
| FREEDRAG | 0.90 | 32.30 | 30.37 | 51.42 | 0.07 | 200 | 300 | OPTIM |
| DRAGTEXT | 0.87 | 32.87 | 29.59 | 42.93 | 0.12 | 80 | 80 | OPTIM |
| ADAPTIVEDRAG | 0.86 | 35.70 | 32.94 | 77.73 | 0.07 | 80 | 300 | OPTIM |
| CLIPDRAG* | 0.88 | 32.30 | / | / | 0.07 | 160 | 160 | OPTIM |
| GDRAG* | **0.91** | 26.49 | / | 152 | 0.08 | 80 | 250 | OPTIM |
| DRAGLORA | 0.87 | **23.77** | 22.70 | 29.84 | 3.19 | 80 | 80 | OPTIM |
| DRAGLORA-FAST | 0.88 | 27.55 | 26.90 | **22.89** | 3.19 | 80 | 80 | OPTIM |
| INSTANTDRAG | 0.88 | 31.51 | 28.67 | **1.44** | 914 | 0 | 0 | ENC |
| LIGHTNINGDRAG | 0.89 | 29.10 | 26.77 | 1.95 | 933 | 0 | 0 | ENC |
| SDE-DRAG | **0.91** | 44.48 | 41.53 | 62.74 | 0 | 100 | 0 | TRFREE |
| FASTDRAG | 0.86 | 31.62 | 28.29 | 5.13 | 0 | 0 | 0 | TRFREE |

cant edits, it fails to reflect the consistency. Following (Ling et al., 2024), we take a drag-back pipeline to simultaneously measure editability and consistency. After one round of editing, we train a reconstruction LoRA on the edited image, swap the source points and target points, and perform a second round of drag editing. The discrepancy between the second-edited image and the original image is measured by LPIPS and CLIP (Radford et al., 2021). The CLIP metric calculates the similarity of features extracted by the CLIP image encoder from two images. As shown in Figure 5 and Table 2, lower LPIPS and higher CLIP indicate that both rounds of drag editing effectively preserve the original image information, and also suggest that the first round of editing brings the image sufficiently close to the target. Our DragLoRA achieves better drag-back results both visually and quantitatively.

*Table 2.* Drag-Back evaluation on DragBench.

| METHODS | LPIPS (x10) ↓ | CLIP ↑ |
|---|---|---|
| DRAGDIFFUSION | 1.39 | 0.96 |
| DRAGNOISE | 1.64 | 0.94 |
| DRAGLORA | **1.33** | 0.96 |
| GOODDRAG | 1.37 | **0.97** |

### 4.4. Ablation Study

To systematically evaluate the contributions of different modules to the overall performance, we conduct an ablation study, starting from the baseline and incrementally adding modules. We assess their impact on performance metrics 1-LPIPS and MD. The results are summarized in Table 3.

Our baseline starts by optimizing LoRA instead of input latent feature $z_{35}$, based on DragDiffusion. Due to the excessive updates of LoRA, distortive changes occur in the image, resulting in poor performance. Applying DOO effectively stabilizes the drag updates, improving both LPIPS and CLIP. ILFA helps strengthen the edit accuracy by aligning the layout information of input features with LoRA weights. The EPT and ASS are primarily designed to enhance efficiency. The former mitigates the interference from error-tracked points, while the latter makes the training process more adaptive to diverse scenarios, thereby also improving editability. More visible results can be found in Appendix B.4.

*Table 3.* Ablation study on DragLoRA.

| METHODS | 1-LPIPS ↑ | CLIP ↑ | MD ↓ | M-MD ↓ |
|---|---|---|---|---|
| BASELINE | 0.87 | 9.47 | 48.55 | 41.00 |
| + DOO | **0.93** | **9.8** | 36.88 | 34.21 |
| + ILFA | 0.88 | 9.74 | 26.99 | 26.78 |
| + EPT | 0.88 | 9.74 | 25.45 | 25.24 |
| + ASS | 0.87 | 9.73 | **23.77** | **22.70** |

## 5. Conclusion

We introduce DragLoRA, a novel framework for drag-based image editing that improves precision and efficiency through online optimization of LoRA adapters. By replacing latent feature optimization with dynamic model adaptation, DragLoRA enables finer deformations while preserving semantic fidelity. The dual-objective optimization, combining drag loss and DDS loss, ensures alignment with pretrained diffusion priors, addressing instability from unrestricted LoRA

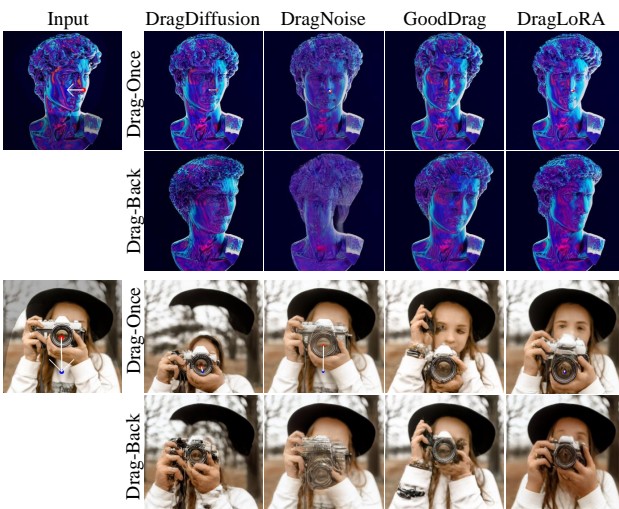

Figure 5. Comparisons in Drag-Back pipeline. Following (Ling et al., 2024), we perform two symmetric drag edits: the first adheres to the input drag annotation and the second reverses it. We focus on the similarity between the drag-back image and the input to validate image fidelity and edit accuracy.

fine-tuning. The cyclic input feature adaptation and adaptive optimization further stabilize motion supervision and boost efficiency. Experiments show DragLoRA outperforms existing methods in both precision and runtime, making it a powerful tool for interactive image editing. Future work will extend this framework to support flexible drag tasks, including region-based drag, incorporating reference images, and various types of drag adaptation.

**Limitations**. Our work has several limitations that we leave for the future work. First, considering the quantitative metrics are not direct and accurate, we plan to conduct a user study in the future. Second, there are challenging cases where DragLoRA does not perform optimally, producing edited image with low fidelity, such as moving the camera down to show up the face behind it, which also challenges other methods. We believe advanced generative models can be applied to improve the quality of edited images.

## Acknowledgements

This work was supported by the Science and Technology Commission of Shanghai Municipality under Grant No.22511105800, 19511120800 and 22DZ2229004, the AI project from the economic and information commission of shanghai (Grant No. 2024-GZL-RGZN-01038), and ECNU Multifunctional Platform for Innovation (001).

## Impact Statement

This paper presents work whose goal is to advance the field of Machine Learning. There are many potential societal consequences of our work, none which we feel must be specifically highlighted here.

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

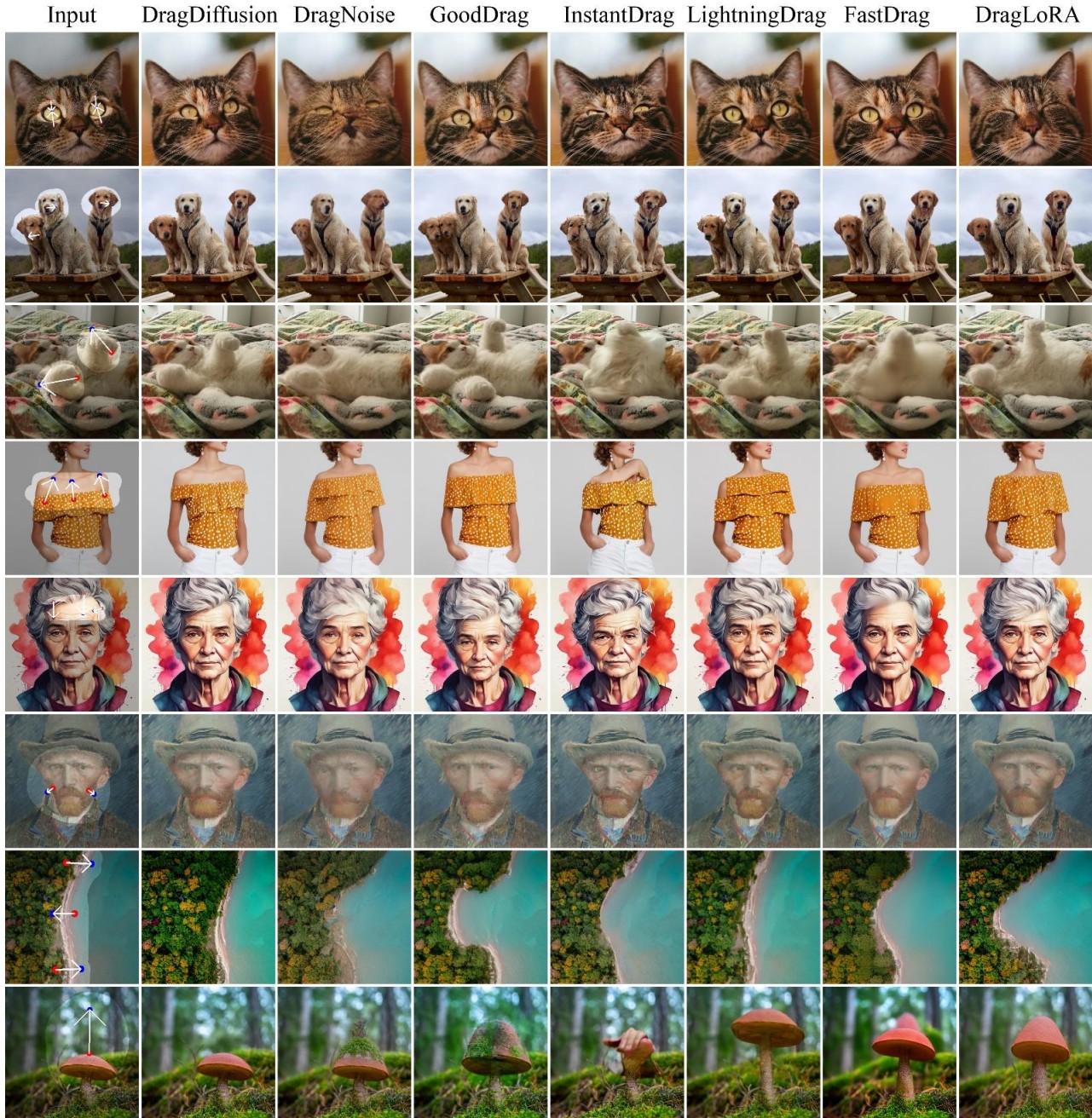

*Figure 6.* Qualitative comparisons with more methods (Shi et al., 2024b; Liu et al., 2024; Zhang et al., 2024; Shin et al., 2024; Shi et al., 2024a; Zhao et al., 2024).

## A. Additional Results

More qualitative comparison results are given in Figure 6. It can be observed that DragLoRA achieves the most accurate drag-edit results. As evidenced by Figure 7, the minD of DragLoRA averaged on 205 images from DragBench reaches the lowest through 80 steps, which demonstrates the stability and reliability of our proposed optimization process. Notably, minD-related strategies (ASS and EPT) are not designed to decrease minD, and minD curve comparisons between DragLoRA and other methods are fair. When ASS and EPT are ablated (DragLoRA-wo/minD, dark purple), the observed minD is even lower than DragLoRA. To further evaluate the efficiency of point displacement, we compute the Euclidean distance between

handle points and target points $d(\mathbf{h}_i^{k+1}, \mathbf{g}_i)$ at each drag step, and report the value averaged over all points and images, which is noted as dT. Results in Figure 7 confirm that DragLoRA drives handle points to target positions more efficiently than various existing methods.

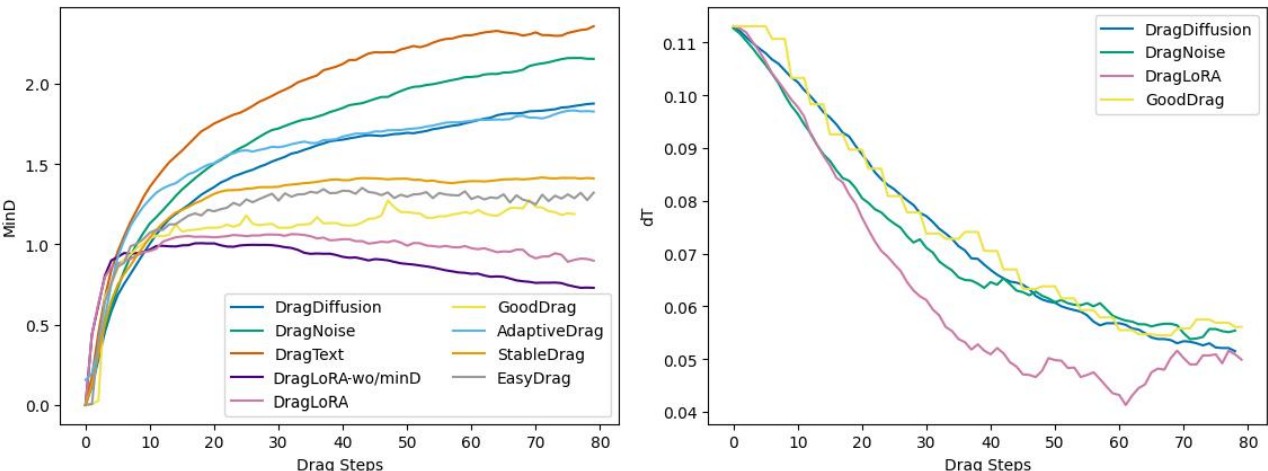

*Figure 7.* Comparisons of $minD$ and $dT$ across different methods. For each benchmark image, we record $minD$ and $dT$ from point tracking after each optimization step and compute the average curve across all images. DragLoRA achieves the lowest $minD$ and $dT$, indicating superior dragged results.

## B. More discussion

In the main paper, we have studied the effects of Dual-Object Optimization (DOO), Input Latent Feature Adaptation (ILFA), Adaptive Switching Scheme (ASS) and Efficient Point Tracking (EPT). In this section, we provide more analysis on DOO, ILFA and details bout EPT. Furthermore, we provide a qualitative ablation study as a supplement to the quantitative results.

### B.1. Dual-Object Optimization

In this section, we analyze the difference between the loss introduced in (Hertz et al., 2023) and our $L_{\text{DDS}}$.

In the original DDS paper, $\nabla_\theta L_{\text{DDS}} = (\epsilon^{edit} - \epsilon^{ori})\frac{\partial z_0}{\partial \theta}$ is applied to guide the image towards the semantic editing direction, which points from the original text to the target text. In our work, however, we employ $L_{\text{drag}}$ for the editing operation and use $\nabla_{\Delta\theta} L_{\text{DDS}} = (\epsilon^{ori} - \epsilon^{drag})\frac{\partial \tilde{z}_0}{\partial \Delta\theta}$ as a regularization term. This regularization constrains the edited model to remain close to the original model's generative capability, thereby preventing excessive updates. In essence, our application of the DDS loss is intentionally reversed relative to the original purpose. Therefore, the backpropagated gradient in our method carries an additional negative sign compared to the original DDS loss function.

### B.2. Input Latent Feature Adaptation

In this section, we analyze the necessity and theoretical feasibility of ILFA discussed in Section 3.3, building a connection with Score Distillation Sampling (SDS) (Poole et al., 2022) and delta denoising score (DDS) (Hertz et al., 2023). Furthermore, we apply ILFA to DragNoise (Liu et al., 2024) to validate its generalization.

**Necessity**. Maintaining a fixed input feature $z_{35}$ while solely optimizing DragLoRA preserves the initial spatial layout. This constraint forces the LoRA to generate the growing displacements as handle points approach their targets, creating a fundamental conflict with the small-step motion supervision paradigm. As visualized in the cross-attention maps (Figure 8), this mismatch manifests as unclear outline of duck's beak and suboptimal editing results. To resolve this instability, we propose Input Latent Feature Adaptation (ILFA), which dynamically aligns $z_{35}$ with the LoRA-learned motion through a denoise-renoise update strategy.

**Formula Analysis**. In ILFA, we denoise the input latent feature $z_t, t = 35$ to $z_{t-1}$ by Equation (2) and then renoise it to

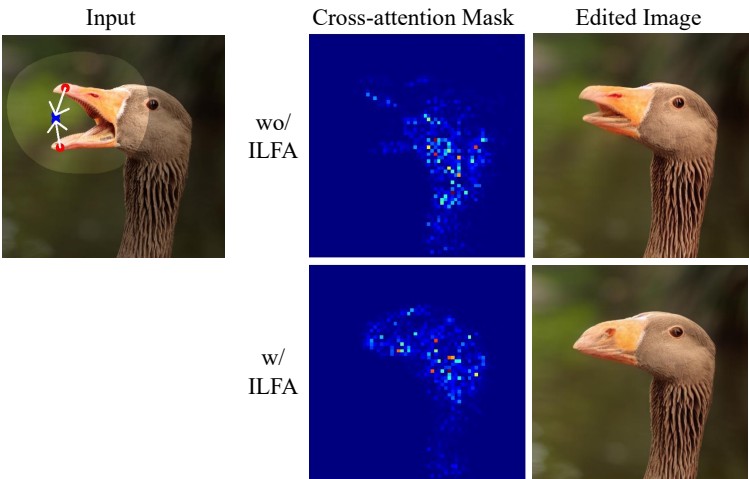

*Figure 8.* Visualization of cross-attention map to reveal layout conflicts between the fixed input latent feature and DragLoRA. ILFA mitigates these conflicts and produces more precise editing results.

obtain the new $z_t$ by Equation (1). We can combine the two equations and rewrite them into a single form as:

$$z_t = \sqrt{\alpha_t} \left( \sqrt{\bar{\alpha}_{t-1}} \left( \frac{z_t - \sqrt{1 - \bar{\alpha}_t} \, \epsilon_{\theta + \Delta\theta}(z_t, t)}{\sqrt{\bar{\alpha}_t}} \right) + \sqrt{1 - \bar{\alpha}_{t-1}} \cdot \epsilon_{\theta + \Delta\theta}(z_t, t) \right) + \sqrt{1 - \alpha_t} \cdot \epsilon_t$$

$$= z_t + \sqrt{1 - \alpha_t} \cdot \epsilon_t - (\sqrt{1 - \bar{\alpha}_t} - \sqrt{\alpha_t - \bar{\alpha}_t}) \cdot \epsilon_{\theta + \Delta\theta}(z_t, t) \tag{9}$$

Essentially, ILFA involves taking the weighted difference between the DragLoRA prediction and random noise, and then using this difference as the editing direction to guide input feature. This is similar to SDS, which takes the difference as grad to optimize the input feature. If we renoise $z_{t-1}$ with DDIM inversion as Equation (4), the guidance form looks like DDS. As shown in Table 4, the SDS form works better in DragLoRA. We attribute this to the fact that random noise is more capable of altering the inherent layout information and creating suitable information that needs to be filled in.

*Table 4.* Comparison on ILFA forms.

| METHODS | 1-LPIPS | MD |
|---|---|---|
| ILFA-DDS | 0.91 | 29.33 |
| ILFA-SDS | 0.87 | 23.77 |

**Generality**. ILFA is portable and can be applied to other methods like DragNoise, which optimizes the intermediate feature of Unet. As shown in Figure 9, ILFA improves the editability and reduces the ambiguity caused by layout conflicts between the input and the intermediate layer.

### B.3. Efficient Point Tracking

As introduced in Section 3.5, the common neighborhood region is so large that the tracked points could be misled by ambiguous points, causing dragging to be stuck. To improve efficiency and eliminate error-prone reverse-direction points, we tested distance-closer (Jiang et al., 2024), angle-closer and linear (Ling et al., 2024) region on DragLoRA, with a decreasing number of candidate points. The stricter the constraints on the candidates, the more likely the tracked points are to move towards the targets, which may ultimately lead to under-optimization. Therefore, we introduce an additional confidence check. Specifically, when $minD > d_2$, we determine that the tracked point cannot accurately reflect the current editing state, so we retain the previous point coordinates. As shown in Table 5, the Distance-closer method achieves the best drag edits while the other two cost less time. Since we combine DragLoRA with linear point searching in the FreeDrag way, we also compare it with FreeDrag and find that DragLoRA surpasses FreeDrag in every aspect.

Input  DragNoise  +ILFA

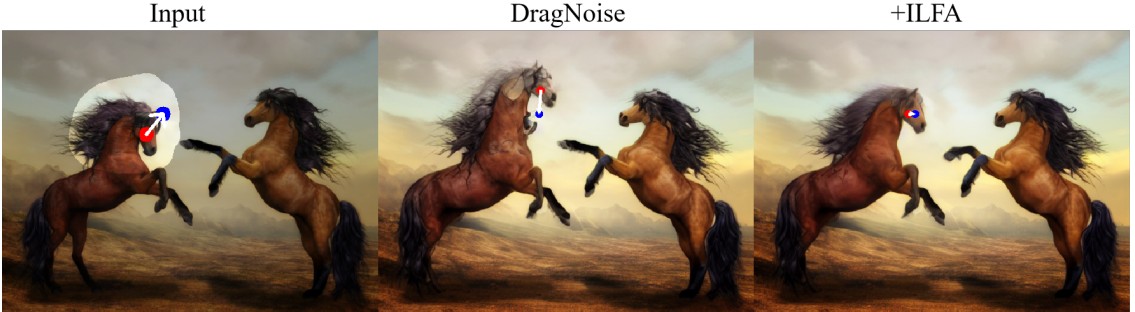

*Figure 9.* Comparison on DragNoise-based ILFA.

*Table 5.* Comparison on strategies in EPT.

| METHODS | 1-LPIPS ↑ | MD ↓ | M-MD ↓ | TIME(S) ↓ |
|---|---|---|---|---|
| DISTANCE-CLOSER | 0.87 | **23.77** | **22.70** | 29.84 |
| ANGLE-CLOSER | 0.88 | 27.55 | 26.90 | 22.89 |
| LINEAR | **0.90** | 30.61 | 30.21 | **20.79** |
| FREEDRAG | **0.90** | 32.30 | 30.37 | 51.42 |

## B.4. Ablation Study

As shown in Figure 10, simply optimizing LoRA accroding to the strategies of DragDiffusion leads to unexpected extreme edits. We increase the image fidelity by applying DOO to restrict the parameters of DragLoRA from deviating largely. As we successively incorporate ILFA, ASS and EPT, DragLoRA significantly enhances the degree of editing, and these significant edits are stable and reliable.

Input  Baseline  +DOO  +ILFA  +ASS & EPT

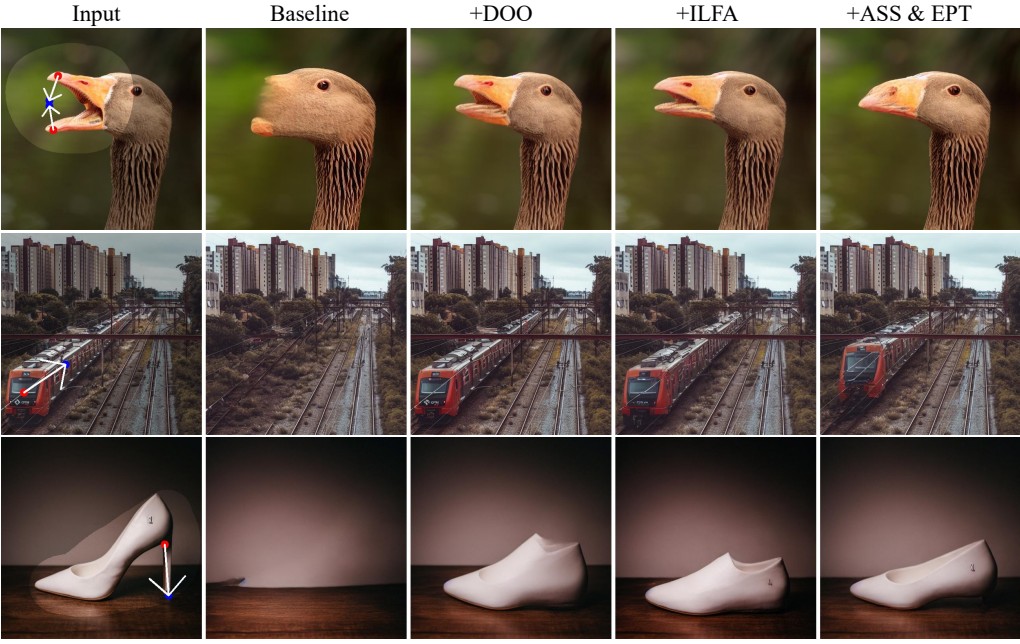

*Figure 10.* Visual ablation study.

