# OpenReview forum: "DragLoRA: Online Optimization of LoRA Adapters for Drag-based Image Editing in Diffusion Model"
_ICML.cc/2025/Conference — ICML 2025 poster_

### Official Review · Reviewer_Bmt5 · 2025-03-11

**Overall Recommendation:** 3

**Summary:**

The paper proposes a new method, named DragLoRA, for drag-based image editing using a pre-trained Stable Diffusion model. Specifically, the paper proposes two novel steps: DragLoRA online optimization (DOO) and Input Latent Feature Adaptation (ILFA). In DOO, instead of optimizing the latent representation like the classic DragDiffusion, it optimizes the LoRA with feature drag loss and the DDS loss. In IFLA, the latent representation passes through DragLoRA and is renoised to adapt the latent representation to dragging. Meanwhile, the paper also discovered that IFLA itself could occasionally achieve point moving, even without DOO steps. Therefore, the paper proposes an adaptive switching scheme (ASS) based on point tracking: if IFLA itself can achieve reliable point moving, DOO can be skipped for this step until the result of point tracking reveals that IFLA cannot provide a good move, DOO is resumed to supervise point moving using optimization. DragLoRA is evaluated on the DragBench benchmark against previous methods and shows superior performance in the MD metric with comparable computational time.

## Update after rebuttal
I appreciate the author's response, and it solved most of my concerns. Therefore, I keep my original rating.

**Claims And Evidence:**

Based on the results of experiments, DragLoRA improves the baseline on MD and m-MD metrics in the DragBench benchmark (Table 1). The effectiveness of each proposed components: DOO, ILFA, ASS and EPT are demonstrated by the ablation study in Table 3 and Figure 10 in the supplementary material. Therefore, the effectiveness of DragLoRA could not be denied. However, some points require further clarification to understand how they work, and I lay out my questions regarding this part in **Methods And Evaluation Criteria** section.

Experiments are thorough in general, and they do substantiate the claims. However, I have some concerns and questions regarding the experiments, and I elaborate on these in **Experimental Designs Or Analysis** section.

**Essential References Not Discussed:**

All relevant works have been included in the literature review.

**Experimental Designs Or Analyses:**

1. In Table 1, it is shown that DragLoRA achieves better results than its baseline DragDiffusion with a comparable inference time. Given the fact that DragLoRA has a more complicated pipeline and a larger number of parameters to optimize, I guess that the reason for DragLoRA to catch up DragDiffusion in inference time is ASS, which allows DragLoRA to skip several optimization steps. Therefore, it would be best to show how much time and how many optimization steps can be saved by adopting ASS.

2. The curves that compare DragLoRA and DragDiffusion in Figure 1 and Figure 8 are good. It shows the variation of the quality of the handle point during the optimization process. However, the plot is misleading in terms of drag steps. The plot shows that the DragLoRA finishes editing with fewer drag steps than DragDiffusion, implying DragLoRA is faster and more efficient than DragDiffusion. This is not true according to the results in Table 1, which shows that DragLoRA and DragDiffusion have similar inference times. According to Algorithm 1, DragLoRA only increment drag step count $i$ when DOO is triggered. All steps in the while loop are ignored, which makes DragLoRA appear to be much more efficient. A more appropriate curve would be that the drag step still counts ILFA in the while loop but uses a different color for this section, indicating that these steps are faster than the optimization step. This can also reflect how many optimization steps are skipped.

3. Figure 7 in the supplementary material plots MinD vs drag steps for all methods. It would be better if $d(h_{i}^{k}, t_{i})$ vs drag steps can also be plotted in a similar way as it will reflect how fast each method completes the dragging.

4. Although mean distance is widely used to evaluate the dragging result, it cannot always reflect the true quality of the editing. A more appropriate metric is the human evaluation, which is widely adopted in text-to-image and text-to-video generation methods. However, it is inappropriate to ask for a human evaluation during the author-reviewer discussion period, and I simply want to share my opinion on it.

**Methods And Evaluation Criteria:**

Although the effectiveness of the proposed method is empirically demonstrated, further clarification is required on some points:

1. The paper employs the DDS loss to reduce the performance degradation caused by LoRA optimization. The original DSS loss aims to have a cleaner gradient than the SDS loss in editing direction by deducting bias from the editing SDS loss. Mathematically, it is $\nabla_{\theta} L_{DSS} = \nabla_{\theta} L_{SDS}(\hat{z}, c_{edit}) - \nabla_{\theta} L_{SDS}(z, c_{ori})$. The bias is $\nabla_{\theta} L_{SDS}(z, c_{ori})$ and the intuition is that if the latent of the original image $z$ already aligns with the text $c_{ori}$, the derivative should be zero and any remaining value is bias. If we apply the analogy to this case, shouldn't the DSS loss be $(\epsilon^{drag} - \epsilon^{ori})\frac{\partial \hat{z_{0}}}{\partial \Delta \theta}$ rather than $(\epsilon^{ori} - \epsilon^{drag})\frac{\partial \hat{z_{0}}}{\partial \Delta \theta} $ as shown in equation 8?

2. The adaptive optimization scheme noticed that after DOO, LoTA can move the handle point toward the targets without DOO. There lacks a intuition behind this phenomenon. Finetuning LoRA for one moving step is very likely to cause LoRA to overfit to that specific step, so it is quite counter-intuitive in this point.

**Other Comments Or Suggestions:**

There are several minor issues:

1. In the right column of Ln200, "we adopt the initla latent input $z_{35}^{0}$" should be "we adopt the initla latent input $z_{35}^{0} = z_{35}$"

2. In Section 4.1 Implementation Details, $\lambda_{Mask}$ and $\lambda_{DSS}$ are not defined in the previous section.

**Other Strengths And Weaknesses:**

All strengths and weaknesses are laid out in the previous sections.

**Questions For Authors:**

I would appreciate it if the authors could address my concerns regarding the Method and Experiment sections.

**Relation To Broader Scientific Literature:**

The paper addresses the drag-based image editing problem using diffusion models. It is built on top of DragDiffusion, which is one of the pioneering works in this direction. The general pipeline of this work is similar to that of DragDiffusion, and it proposes its own innovation to improve the baseline.

**Theoretical Claims:**

This paper is an application paper with no significant theoretical contribution.

---

> ### Author Rebuttal · Authors · 2025-03-30
>
> Thank you very much for your insightful and constructive feedback. We appreciate the depth of your analysis and the valuable suggestions you provided. Below, we address your concerns point by point:
>
> ### **1. Difference from the Original DDS Loss.**
>
> In the original DDS paper, $\nabla_{\theta} L_{\text{DDS}}=(\epsilon^{edit}-\epsilon^{ori})\frac{\partial{z_0}}{\partial {\theta}}$ is applied to guide the image towards the semantic editing direction, which points from the original text to the target text. In our work, however, we employ $L_\text{drag}$ for the editing operation and use $\nabla_{\Delta\theta} L_{\text{DDS}}=(\epsilon^{ori}-\epsilon^{drag})\frac{\partial{\hat{z_0}}}{\partial {\Delta\theta}}$ as a regularization term. This regularization constrains the edited model to remain close to the original model’s generative capability, thereby preventing excessive updates. In essence, our application of the DDS loss is intentionally reversed relative to the original purpose. That's why we apply grad $\epsilon^\text{ori} - \epsilon^\text{edit}$ to $\hat{z}_0$.
>
> ### **2. Intuition Behind ILFA Without DOO.**
>
> We observe that after sufficient DOO (Dual-objective LoRA optimization), ILFA (Input Latent Feature Adaptation) can solely move the handle point. In this setting, accumulated DOO gradients from previous optimizations, can be utilized for moving handle points at the new positions without extra driving force. In each gradient step, although the specific tasks are not exactly the same, they share a low-variance handle feature and a common direction. DragLoRA can learn these commonalities and generalize, which is comparable with meta-learning. Note that ILFA-only is not stable enough, so we introduce adaptive switching scheme (ASS). Here we explain this process and intuition behind this:
>
> * **Initialization and Thresholding:** We initialize DragLoRA from a reconstruction LoRA and activate the no-training mode only after LoRA has been optimized for a certain number of steps. This ensures that the LoRA module learns a robust directional signal over multiple iterations, rather than merely shifting from one fixed point to another.
> * **Optimization Confidence Assessment:** We employ minD and coordinate distance of points to evaluate the confidence of previous-step LoRA optimization. When LoRA's guidance is deemed reliable with an effective movement and a smaller value of minD, ILFA can incrementally adjust the nearby handle points in a small, controlled manner, thereby effectively advancing in the intended direction.
> * **Fallback Mechanism:** In cases where LoRA is compromised due to overfitting, ILFA can hardly advance the handle points that LoRA is unfamiliar with or produce large minD. Our ASS strategy can detect these unreliable cases and switch back to DOO with a new supervisory signal.
>
> In summary, IFLA-only can be potentially used with detection metric as the safty guard.
>
> ### **3. Time Efficiency via Adaptive Switching.**
> You correctly noted that our ASS allows us to bypass several DOO steps by relying on the training-free ILFA mechanism. This substantially reduces the number of optimization steps and, as a result, saves significant time. Specifically, the time consumed by one DOO step is around **0.26s** while one ILFA step costs around **0.05s**, and we apply **54** DOO steps and **87** ILFA steps per image on DragBench.
>
> ### **4. Visualization of the Optimization Process.**
> We appreciate your suggestion to enhance the visualization of our optimization process. In response, we have supplemented our manuscript with two additional figures that more clearly illustrate the progression of the drag editing process. The first image, which separates the ILFA-only steps (displayed in a different color) from the DOO+ILFA steps, clearly indicates how many optimization steps are skipped. And the second image, which calculates the average normalized Euclidean distance $d(h_i^k,t_i)$ on DragBench, indicates how fast the handle points move. You can view these visualizations at the following link: https://imgur.com/a/PVuceIo.
>
> ### **5. Incorporating Human Evaluation.**
> We agree that human evaluation is crucial for fully assessing the quality of editing results. While our current work primarily relies on quantitative metrics, a comprehensive user study would provide further insights. We plan to conduct such evaluations and release the results in our Github.
>
> ### **6. Symbol Corrections and Minor Issues.**
> We have noted the suggested corrections and will address them in the revised version of our manuscript.
>
> Thank you once again for your detailed and thoughtful comments. We eagerly await your feedback on our rebuttal.

---

> > ### Comment · Reviewer_Bmt5 · 2025-04-04
> >
> > Thank you for your detailed response. It addresses all my concerns. I am particularly grateful that the author illustrates the trend of DOO and ILFA along the editing. I have no more questions.

---

> > > ### Author Response · Authors · 2025-04-09
> > >
> > > Thank you once again for your insightful questions and valuable suggestions. We deeply appreciate your expertise, which has been instrumental in helping us improve our submission.

---

### Official Review · Reviewer_TdyS · 2025-03-12

**Overall Recommendation:** 3

**Summary:**

The authors presented the DragLoRA method for efficient and more accurate drag-editing, which is specified by a mask of the area to be edited and pairs of points specifying the direction of shift editing. The method is based on the use of LoRA adaptors for simultaneous gradual shifting towards target points, but at the same time keeping the model prediction with adaptors close to the original model ones. For this purpose, the authors use dual-objective LoRA optimization (DOO), combining feature shift loss and DDS loss. In addition, the authors propose an adaptive switching scheme (ASS), the idea of which is to choose a strategy with or without updating the adapter weights depending on how far the handle point is from the target point. The authors also propose Input Latent Feature Adaptation (ILFA) mechanism, which is to renoise the latent once by 1 step and then noise with Gaussian noise. The authors do this adaptation at each step of the algorithm regardless of whether there is an optimization step, and the goal is to ensure coherent updates on LoRA parameters across iterations. Finally, the authors use Efficient Point Tracking (EPT) to efficiently select the next handle points, reducing the complete enumeration. The proposed method requires fewer optimization steps compared to existing methods and preserves the details of the original image better.

**Claims And Evidence:**

The authors provide an extensive comparison with existing methods, both quantitative and qualitative. However, there are questions about the visual quality of the method. In Figure 4, the first and third rows show results that are better than the baselines, but the proposed method outputs a rather unrealistic image. The girl in the first row has unrealistic facial features and the camera has lost its shape. Given that these images are highlighted as the best, there are questions about the real quality of the method and how applicable it is in practice.

**Essential References Not Discussed:**

GDrag [1] is one of the latest published methods that shows high metrics on a drag-editing task.

The SDE-Drag [2] article showed that using SDE instead of ODE improves the quality of various types of editing, including drag-editing.

[1] GDrag:Towards General-Purpose Interactive Editing with Anti-ambiguity Point Diffusion, Xiaojian Lin, Hanhui Li, Yuhao Cheng, Yiqiang Yan, Xiaodan Liang

[2] The Blessing of Randomness: SDE Beats ODE in General Diffusion-based Image Editing, Shen Nie, Hanzhong Allan Guo, Cheng Lu, Yuhao Zhou, Chenyu Zheng, Chongxuan Li

**Experimental Designs Or Analyses:**

For the experimental part of the work, the authors chose to compare the proposed method with baselines on the DragBench dataset. The metrics chosen for quantitative comparison are 1-LPIPS, Mean Distance (MD) and masked Mean Distance. Also, a comparison in terms of the running time and the number of parameters is added in Table 1. From the comparison, it is observed that in terms of MD and m-MD metrics, the proposed method is indeed on top. However, I think it is not very fair to compare by runtime. The authors write in Section 4.3 that for DragLoRA, only the time required for finetuning for editing was included in the comparison table, and the time for initializing this finetuning (actually finetuning on reconstruction) was not included. In this regard, there is a question, for the DragDiffision method the corresponding time was also excluded in the comparison? What is the order of the time required for this step in general? Is it possible to neglect it in the comparison on the basis of the fact that it can be done offline, although the user still has to wait for this time to upload his image?

The authors also offer an interesting analysis on the Drag-Back Pipeline. The image is first edited with a set of point pairs and then edited again in the reverse direction of point shifts. This should ideally result in a reconstruction of the original image. The authors compare a subset of methods on this task and show in Table 2 that in terms of LPIPS and CLIP metrics, which are responsible for image preservation, the proposed method performs almost the best, which is supported by the visual comparison in Figure 5.

Finally, the authors conducted an ablation study (Table 3) in which the authors show the need for each of the four components of the method, which was supported by the visual comparison in Figure. 10.

**Methods And Evaluation Criteria:**

The proposed method solves the drag-editing problem, which has been widely spread recently, as it could simplify the image editing process for real users. The problem has been studied many times and methods that solve it have been proposed, and there are also benchmarks such as DragBench in order to compare the methods proposed in the community. The authors use just this benchmark for comparison. The metrics that the authors use are common and standard for this task.

**Other Comments Or Suggestions:**

Typos:

127: InstanceDrag → InstantDrag

305: DragDiffuaion → DragDiffusion

660: Aadption → Adaption

**Other Strengths And Weaknesses:**

**Strengths:**

A clear and reasonable idea of the method.

**Weaknesses:**

The method is based on the outdated sd-1.5 model, the quality of which is initially inferior to such models as SD3, Flux.

**Questions For Authors:**

1. My main question is, as I wrote earlier about measuring the runtime of the method. For the DragDiffision method the corresponding time was also excluded in the comparison? What is the order of the time required for this step in general? Is it possible to neglect it in the comparison on the basis of the fact that it can be done offline, although the user still has to wait for this time to upload his image?
2. How were the constants used in the pseudocode selected? What is their intuition? How did they get out? Is there an analysis of them?
3. How do you assess the applicability of the proposed method, taking into account such significant distortions in the visual comparison in Figure 4, as I wrote earlier? Will the user be satisfied with such unrealistic output images?

**Relation To Broader Scientific Literature:**

The idea of optimizing weights by gradually shifting points in a given direction is not new and has been proposed as early as DragGan and DragDiffusion. However, the authors proposed a number of modifications that improved the quality of the Pipeline. As mentioned earlier, the proposed method consists of 4 main components.

1. Dual-objective LoRA optimization (DOO)

2. Input Latent Feature Adaptation (ILFA)

3. Adaptive switching scheme (ASS)

4. Efficient Point Tracking (EPT)

DOO uses delta denoising score, which has been proposed earlier but not used as a component of loRA for optimization in drag-editing. The combination of the two losses is rather a new approach for this task.

ILFA has also not been presented in the literature before, but there was an attempt to use a close idea in GoodDrag.

ASS is an entirely new idea, although it stands out for its simplicity.

Finally, EPT has already been proposed in papers such as CLIPDrag and FreeDrag and is not novel.

Overall the combination of these ideas has shown to be quite good, there are both completely new ideas as well as utilizing existing ones as separate components. The main novelty is in the combination of these parts with each other, which gave a fairly high quality output.

**Theoretical Claims:**

For the most part, there is no theoretical evidence in the article. However, the supplementary contains the derivation of formula 9 for latent updating using the Input Latent Feature Adaptation (ILFA) mechanism. I see no inaccuracies in this conclusion.

---

> ### Author Rebuttal · Authors · 2025-03-30
>
> Thank you for your thorough review and constructive feedback. Below are our point-by-point responses:
>
> ### **1. Runtime Comparison Fairness.**
> Our reported time excludes the offline LoRA finetuning time for reconstruction (~48s per image over 80 steps on NVIDIA 4090 GPU), which applies to **all methods** with reconstruction steps $>0$.
> We agree that this is long and it can't be ignored. However, the trained LoRA can be stored and reused for multiple drag editings by various annotations of points. They can also be applied in other customized editing. Here, the exclusion of offline training emphasizes the comparison of time cost on dragging. We will describe this cost in the revised paper.
>
>
> ### **2. Clarification on minD.**
> We propose minD, which measures the distance between feature of the tracked new handle points and initial handle point, to serve as a training-time indicator of the stability of the LoRA optimization. However, lower minD does not definitly result in good drag quality. Metrics such as MD and m-MD are more reliable for evaluating the final results. Therefore, minD curves are used only for reference.
>
> Specifically, minD is applied in Adaptive Switching Scheme (ASS) and Efficient Point Tracking (EPT). In ASS, minD and coordinate distance assess whether LoRA or input latent feature updates are confident. If updates are confident, the system enters IFLA-only mode; otherwise, $L_{drag}$ continues driving in next step. In EPT, minD checks if tracked handle points accurately reflect the previous updates. If minD is low, the system moves to the next step; otherwise, it stays with additional motion supervision. MinD does not decrease in ASS, and is only lowered down in EPT. When the ASS and EPT strategies are ablated (DragLoRA-wo/minD, dark purple), as shown in https://imgur.com/a/UYhzqfD, the observed minD is even lower than DragLoRA. So we mainly use minD to gauge the online optimization, but do not reduce its value in purpose. The lower minD in DragLoRA mainly comes from the LoRA structure and DOO.
>
> ### **3. Incorporation of GDrag and SDE-Drag.**
> Thank you for the advices on adding essential references. SDE-Drag presents a unified probabilistic formulation for diffusion-based image editing, including drag, while GDrag categorizes point-based manipulations into three atomic tasks with their dense trajectory, and jointly optimizes point-wise adaptation scales and latent feature biases at a sampled timestep, achieving less ambiguous outputs.
>
> We adopt the metric values reported by GDrag and evaluate SDE-Drag on DragBench introduced by DragDiffusion. As shown below, DragLoRA achieves better editing accuracy than the others while GDrag achieves better image consistency.
>
> | Methods        | 1-LPIPS↑ | MD↓    | m-MD↓  | Time(s)↓ | Params(M) | RecSteps | DragSteps | Category |
> |----------------|----------|--------|--------|----------|-----------|----------|-----------|----------|
> | SDE-Drag       |  **0.91**   |  44.48 | 41.53 | 62.74|     0     | 100        |     0     | TrFree   |
> | GDrag         | **0.91** |  26.49 |   /   |     /    | 0.08      |  80      | 250       | Optim    |
> | **DragLoRA**   | 0.87     | **23.77**|  **22.70** | **29.84** | 3.19      | 80       | 80        | Optim    |
>
> ### **4. Generative model Limitation.**
> Since all methods are based on Stable Diffusion 1.5, we also implement our method on this model for fairness. While Flux/SD3 have higher generative capacity, indicating the potential to boost editing performance, they require more cost on both time and computation sources. We will explore this potential in the future and release code for easy model swapping among SDXL and Stable Diffusion 2.1.
>
> ### **5. Parameter Selection in Pseudocode.**
> Since focusing on practical strategy, we do have some threshold params in Alg. 1 determined via empirical analysis: $k_{ini}=10,th_{low}=1,th_{high}=1.3,d_m=1.4$. As shown in Fig.7, after $k_{ini}$ steps, minD plateaus, indicating that LoRA optimization tends to be stable. According to the observation on DragLoRA wo/ ASS, minD $>th_{high}$, usually correlates with unrealistic edits. Besides, we set $minD<th_{low}$ and $||h-n||_2<d_m$ to indicate high confidence to balance the efficiency and stability. However, these two values are still not fully tuned.
>
> ### **6. Current Limitations of Drag-Editing Methods.**
> We acknowledge that a limitation common to current drag-editing methods, including our own, is that in some challenging cases the output may still appear unnatural or low edit-accuracy. In the case of "moving the camera away to supplement the girl's facial details," the editing results can be suboptimal. We believe that incorporating reference face ID additionally could help meet user expectations better.
>
> ### **7. Symbol Corrections and Minor Issues.**
> We have noted the suggested corrections and will address them in the revised version.
>
> Thank you again for your valuable comments. We eagerly await your feedback on our rebuttal.

---

> > ### Comment · Reviewer_TdyS · 2025-04-02
> >
> > Thanks for the comments provided. There is more clarity, especially on the runtime evaluation.

---

> > > ### Author Response · Authors · 2025-04-09
> > >
> > > Thank you once again for your valuable feedback and the time you dedicated to reviewing our work. We greatly appreciate your constructive comments.

---

### Official Review · Reviewer_xAC9 · 2025-03-13

**Overall Recommendation:** 3

**Summary:**

This paper introduces the DragLoRA framework to enhance point tracking in drag-based image editing, thereby improving editing precision. It proposes a DDS loss combined with drag loss, along with a cyclic denoise-renoise process to maintain semantic fidelity with the source image. Additionally, an adaptive optimization scheme is employed to minimize further LoRA adjustments, improving overall efficiency.

**Claims And Evidence:**

All claims are supported by clear evidence.

**Essential References Not Discussed:**

Essential references are all discussed.

**Experimental Designs Or Analyses:**

In the experiments section, the paper selects DragDiffusion, DragNoise, and GoodDrag as the main comparison methods. For DragDiffusion and DragNoise, the same dataset and metrics are used, whereas GoodDrag includes an additional dataset and new metrics. However, these are missing from the quantitative comparison.

**Methods And Evaluation Criteria:**

The method and evaluation criteria make sense for the problem.

**Other Comments Or Suggestions:**

1.	In Figures 2 and 3, some characters appear as "?".
2.	The paper lacks discussion about its limitations.

**Other Strengths And Weaknesses:**

Strength:

1.	The DragLoRA framework enhances the point tracking stage, which is crucial for achieving high image quality and precise drag-editing results.
2.	Various point tracking schemes for drag editing are discussed, and the proposed search strategy is more advanced compared to the vanilla method
3.	In quantitative comparisons, the proposed method is evaluated against many approaches, demonstrating both high image fidelity and time efficiency.

Weakness:

1.	The paper mentions other selected images, more details are needed for this new dataset.
2.	For m-MD, further clarification is required on how it is computed and why it is an improvement over MD.
3.	Based on Table 2, which presents the quantitative comparison on DragBench, the paper claims to achieve state-of-the-art editing quality. However, for 1-LPIPS, STABLEDRAG and EASYDRAG outperform the proposed method, while for m-MD, GoodDrag performs better. In Lines 366–367 of the Evaluation Metrics section, the paper states that m-MD reduces uncertainty errors compared to MD. If m-MD is indeed superior to MD, then the proposed method does not achieve state-of-the-art performance in any metric presented in Table 2.

**Questions For Authors:**

No additional questions.

**Relation To Broader Scientific Literature:**

N/A

**Theoretical Claims:**

No theoretical claims.

---

> ### Author Rebuttal · Authors · 2025-03-30
>
> Thank you for your thoughtful feedback. We address your concerns below:
>
> ### **1. Test data details.**
> We mainly conduct our experiments on DragBench and Drag100, which are proposed by DragDiffusion and GoodDrag. The qualitative comparisons in Fig.4. and Fig.6 are mainly based on these two test sets. The qualitative comparisons on DragBench has been presented in Tab.1 while results on Drag100 are given below:
> | Methods       | MD↓    | m-MD↓  | 1-LPIPS↑ |  Time(s)↓    |
> |---------------|----------|----------|----------|----------|
> | GoodDrag      | 26.72 | 25.26 | 0.86  |  72.06 |
> | DragLoRA      | **26.31**  | **25.20**  | **0.87**|37.93  |
> | DragLoRA-Fast | 29.18 | 27.78 | **0.87** | **24.32**|
>
> Compared to GoodDrag, DragLoRA achieves better MD and m-MD on Drag100, and also reduces editing time by 47.4%, while DragLoRA-Fast reduces editing time by 66.2%.
>
> To enrich the test scenarios, we select a few images from VITON-HD$^{[1]}$ dataset and our private collections, and manually annotate them with handle-target points and masks. Qualitative comparisons on these constructed pairs are presented in row 5 of Fig.4 and rows 3–4 of Fig.6. These original images and results will be released on our Github.
>
> ### **2. MD v.s. m-MD.**
> We observe that applying DIFT to search final handle points $h_f$ around the entire edited image may mislead the found points to excessively deviate from editing region, especially when there are semantically similar points with the source handle in the image, resulting in high MD. E.g., points on the left and right hands are often confused by DIFT. To eliminate these cases, given the user input mask $M$ to indicate the editing region, we treat points within $M$ as candidate set $\Omega_\text{mask}$. Formally, the searching process is:
> \begin{equation}
> h_f = \arg\min_{p \in \Omega_\text{mask}} d(F_{ori}(h), F_{edit}(p))
> \end{equation}
> where $F_{ori}, F_{edit}$ are feature maps extracted by Stable Diffusion 2.1 from original and edited images, $d$ is a cosine distance metric, and $h$ is the given handle points in original image.
> Following MD, we calculate the mean Euclidean distance between target points and their corresponding final handle point to derive m-MD.
> While m-MD is numerically lower than MD, it may yield optimistic scores in cases of image distortion, where the mask constraint forces final handle points closer to the target. Thus, the two metrics complement each other.
> Notably, the inherent limitations of both MD and m-MD affect all methods uniformly, making them fair for comparison.
>
> ### **3. Quantitative comparison.**
> In terms of both MD/m-MD and visual editing quality, our DragLoRA performs comparably to GoodDrag and shows significant advantages over other methods. Meanwhile, our approach achieves the highest efficiency among optimization-based methods, substantially reducing time compared to GoodDrag, which is presented in Tab.1.
>
> Besides, as mentioned in Sec. 4.3, 1-LPIPS inherently favors under-edited results. StableDrag and EasyDrag achieve high 1-LPIPS by insufficiently moving points (see their larger MD/m-MD in Tab.1). As a supplement, we introduce the DragBack pipeline (Sec. 4.5) to measure the similarity between the edited image and original image. After two symmetric edits, there is no doubt that the higher the similarity, the better. As shown in Tab.2, DragLoRA achieves the best LPIPS while GoodDrag achieves the best CLIP, both showing high consistency with original image, which can not be verified in Tab.1.
>
> ### **4. Figure Rendering Issues.**
> Perhaps the "?" symbols in Figs. 2–3 are related to rendering engine version of Edge browser. We reconfirm that:
> symbols render correctly in Chrome, macOS Preview and other PDF viewers.
>
> ### **5. Limitations.**
> We acknowledge that a more detailed discussion of our limitations would improve the manuscript. In the revised version, we will add a section that presents the following points: (1) considering the quantitative metrics are not direct and accurate, we plan to conduct a user study in the future. (2) There are challenging cases where DragLoRA does not perform optimally, producing edited image with low fidelity, such as moving the camera down to show up the face behind it, which also challenges other methods. (3) Due to computational constraints, we have not integrated the latest generative models with our method.
>
> Thank you once again for your valuable comments and suggestions. We eagerly await your feedback on our rebuttal.
>
> [1] Choi S, Park S, Lee M, et al. Viton-hd: High-resolution virtual try-on via misalignment-aware normalization[C]//Proceedings of the IEEE/CVF conference on computer vision and pattern recognition. 2021: 14131-14140.

---

### Official Review · Reviewer_iyEZ · 2025-03-13

**Overall Recommendation:** 3

**Summary:**

This paper introduces DragLoRA, a framework that integrates LoRA into drag-based editing. Instead of optimizing the input feature obtained from DDIM inversion, the paper claims to improve accuracy and efficiency by utilizing lora adaptation. A denoising score distillation loss is proposed to align the outputs with the original model , ensuring stable and accurate features for motion supervision. An adaptive optimization scheme enhances both precision and computational efficiency. The experiments show that DragLoRA boosts efficiency in drag-based editing benchmarks.

**Claims And Evidence:**

yes

**Essential References Not Discussed:**

The paper has discussed essential references

**Experimental Designs Or Analyses:**

The experimental result is based on a well-established benchmark: DragBench and Drag100.  The paper also conducts a sufficient comparison with other methods.  The ablation study also verifies the effectiveness.

**Methods And Evaluation Criteria:**

The paper proposes three modules: A dynamically lora adapter integrated into UNet for optimizing the editing task, a DDS Loss that claimed by the paper to establish "semantic fidelity" by comparing noise predictions of the original and LoRA-enhanced UNet, and "Adaptive Optimization", which is a strategy that dynamically switches between efficient input updates and motion supervision based on tracking quality.

1. The dynamically lora adapter is simply using lora on Unet for fine-tuning the diffusion model, which seems incremental and naive.
2. The paper proposes three modules with complex design, while I do not see strong motivations to do so.  I personally do not like papers that introduce intricate methods without clear justification.  Furthermore, the experiments do not show that the method significantly overperforms other methods by a notable margin.  I think this field still lacks high-quality data at scale, and fancy tricks or heuristic design can only bring incremental improvements.

**Other Comments Or Suggestions:**

target points "t" and timestep "t" should used different denotation.

**Other Strengths And Weaknesses:**

I also highly recommend to make the Preliminaries section easy-to-understand to help readers who are unfamiliar with Motion Supervision and Point Tracking.

**Questions For Authors:**

N/a

**Relation To Broader Scientific Literature:**

N/A

**Theoretical Claims:**

No theoretical claims given.

---

> ### Author Rebuttal · Authors · 2025-03-30
>
> Thank you for your feedback regarding the clarity of our Preliminaries section and symbolic expressions. We appreciate the opportunity to clarify our approach.
>
> ### **More details about drag-based image editing, particularly its motion supervision and point tracking.**
>
> Drag aims to make the semantic positions of the handle points $\mathbf{h}\in \mathbb{R}^2$ reach targets $\mathbf{g}\in \mathbb{R}^2$ through online optimization, breaking down the journey of $\mathbf{h}$ into small steps. Each step consists of **motion supervision** and **point tracking**:
> * **Motion supervision** relies on $L_\text{drag}$ in eq.(4) to optimize input latent feature $z_{35}$, obtained by DDIM inversion, transforming the region around temporal target points $\mathbf{h} + \mathbf{d}$ into handle points region, where $\mathbf{d}$ means a small displacement, and implicitly filling the handle point region by utilizing the generation capability of model.
> * **Point tracking** locates the new handle points $\mathbf{h}$ on updated deep features in Unet, as defined in eq.(6), which gives a new guidance for next-step motion supervision.
>
> This optimization is performed per image, eliminating the need for a training set.
>
> ### **Difference between DragLoRA and LoRA finetuning.**
>
> For our work, the DragLoRA is online optimized to control the dynamic motion, gradually editing local image features towards the final goal, which is quite different from simply fine-tuning the Unet with LoRA based on a set of images or a single image. Instead, we intend to control the optimization and use LoRA to steer the handle points.
>
> ### **Motivation behind each module.**
> Besides, as shown in Fig.10 and Tab.3, only changing the trainable parameters from $z_{35}$ to LoRA $\Delta \theta$, which serves as our baseline, proves to be unstable. To address this, we devise a series of strategies based on problem analysis:
>
> 1. The primary challenge of optimizing LoRA is that excessive deviation in LoRA parameters can compromise the generative capability of model. Regarding this, we introduce $L_\text{DDS}$ as Eq.(8) to build **dual-objective optimization (DOO)**, which aligns the generative capability of the optimized model with the pre-trained model.
>
> 2. Meanwhile, the fixed input feature $z_{35}$ retains its initial spatial layout, which requires the LoRA to generate the growing displacements for handles as they move towards the targets, and it conflicts with the small step during drag, contributing to instability of optimization. This phenomenon can be observed in cross-attention map visualizations (see Fig. 1 at: https://imgur.com/a/Cb01npK), where the conflicts lead to unclear mask of duck's beak and suboptimal results. To mitigate this, we design **input latent feature adaptation (ILFA)** to align the input $z_{35}$, updating it with the motion learned by LoRA in a denoise-renoise way.
>
> 3. Through experiments, we observe that once LoRA is well optimized, applying multiple ILFA steps alone can facilitate the movement of handle points. Since ILFA is training-free, it is faster than DOO. This suggests the potential benefits of a balanced integration of ILFA and DOO. To achieve this balance dynamically, we propose an **adaptive switching strategy (ASS)** that:
> (1) Solely applies ILFA when LoRA has learned the accurate motion.
> (2) Reactivates LoRA optimization when ILFA alone becomes unreliable.
>
> 4. While ASS enhances motion supervision efficiency, we further propose an **efficient point tracking (EPT)** with limited search region and minD-based retreat, achieving an balance between computational cost and editing effectiveness.
>
> Each module in our design is introduced progressively based on problem analysis and experimental observations. Rather than being arbitrary design choices, they address specific limitations of directly applying LoRA. We hope this clarification provides a stronger motivation for our method and demonstrates the necessity of our contributions.
>
> ### **Comparisons with other works.**
> The combination of modules above achieves an much higher editing accuracy than most of the methods, and our results are comparable with GoodDrag. While GoodDrag optimizes input $z$ over multiple timesteps, sacrificing efficiency, our method costs the least time among optimization-based methods.
>
> Thank you again for your insightful feedback. We eagerly await your feedback on our rebuttal.

---

### Decision · Program_Chairs · 2025-05-01

**Decision:**

Accept (poster)

**Comment:**

The paper introduces an innovative framework for enhancing the efficiency and quality of drag-based image editing using diffusion models. The proposed DragLoRA framework integrates a dynamically LoRA adapter into the UNet architecture, which is designed to fine-tune the diffusion model for specific editing tasks. This adapter, combined with a DDS Loss that ensures semantic fidelity by comparing noise predictions, and an Adaptive Optimization strategy that dynamically adjusts between input updates and motion supervision, represents a significant advancement in the field.

The framework's strength lies in its enhancement of point tracking, which is vital for achieving high-quality and precise drag-editing results. The authors also provide a comprehensive analysis of various point tracking schemes and propose an advanced search strategy that surpasses traditional methods. The quantitative comparisons demonstrate that the proposed method achieves high image fidelity and time efficiency, making it competitive with other state-of-the-art approaches.

While the paper does have some limitations, such as the need for more details on the new dataset and further clarification on the m-MD metric, the overall contributions are substantial. The DragLoRA framework addresses a critical gap in the current methodology by offering a more efficient and precise approach to drag-based editing. The ablation studies and comparative analyses provided in the paper further support the effectiveness of the proposed method. Despite some concerns regarding the experimental setup and comparisons with other methods, the paper's innovative approach and promising results make it a valuable addition to the field of image editing.